# Lindbladian reverse engineering for general non-equilibrium steady states: A scalable null-space approach

Leonardo da Silva Souza[1, 2] and Fernando Iemini[2]

[1]*Instituto Federal de Educação, Ciência e Tecnologia de Mato Grosso,*
*Av Rio Grande do Sul, 2131, St. Industrial, 78640-000 Canarana, Mato Grosso, Brazil*
[2]*Instituto de Física, Universidade Federal Fluminense,*
*Av. Gal. Milton Tavares de Souza s/n, Gragoatá, 24210-346 Niterói, Rio de Janeiro, Brazil*

The study of open system dynamics is of paramount importance both from its fundamental aspects as well as from its potential applications in quantum technologies. In the simpler and most commonly studied case, the dynamics of the system can be described by a Lindblad master equation. However, identifying the Lindbladian that leads to general non-equilibrium steady states (NESS) is usually a non-trivial and challenging task. Here we introduce a method for reconstructing the corresponding Lindbaldian master equation given any target NESS, i.e., a *Lindbladian Reverse Engineering* ($\mathcal{L}$RE) approach. The method maps the reconstruction task to a simple linear problem. Specifically, to the diagonalization of a correlation matrix whose elements are NESS observables and whose size scales linearly (at most quadratically) with the number of terms in the Hamiltonian (Lindblad jump operator) ansatz. The kernel (null-space) of the correlation matrix corresponds to Lindbladian solutions. Moreover, the map defines an iff condition for $\mathcal{L}$RE, which works as both a necessary and a sufficient condition; thus, it not only defines, if possible, Lindbladian evolutions leading to the target NESS, but also determines the feasibility of such evolutions in a proposed setup. We illustrate the method in different systems, ranging from bosonic Gaussian systems, dissipative-driven collective spins and random local spin models.

## I. INTRODUCTION

The quest of estimating maps characterizing the dynamics of quantum systems has significantly increased in the recent years, with both theoretical and experimental advances. The general goal is to infer the underlying dynamical equations that drive a system, given only (full or partial) information about the system properties. Several approaches have been proposed, both for closed [1–20] and open system [22–39] dynamics.

In the simplest scenario, that of closed systems (an idealization of a system perfectly isolated from its environment), the dynamics is fully characterized by its Hamiltonian according to Schrödinger's equation. While in a conventional approach one assumes complete knowledge of the Hamiltonian $\hat{H}(t)$, and the goal is to determine the evolution of the system $|\psi(t)\rangle$, the challenge now is the opposite. That is, given a quantum state $|\psi(t)\rangle$, the goal is to reconstruct the Hamiltonian $\hat{H}(t)$ for which the state is the solution of the Schrödinger's equation, which we denote as *Hamiltonian reverse engineering* (HRE). The most studied case concerns time-independent systems, aiming at the engineering of Hamiltonians for specific ground or excited states [1–10]. The extension to quantum quench protocols and time-dependent Hamiltonians has also been discussed [11–20]. It is worth mentioning some relevant implications of such studies, e.g. understanding the classes of Hamiltonians that could generate tailored many-body correlations on their eigenstates, or generating possible shortcuts to adiabaticity.

Reverse engineering in open systems has also been explored. In this case, however, the dynamics are usually more intricate and harder to solve. A common simplification is to consider a specific class of open dynamics whose system is weakly interacting with the environment and constrained by a Born-Markovian approximation. The effective dynamics of the system can thus be expressed by a Lindbladian master equation ($\mathcal{L}$) [21]. Various strategies have been proposed for a *Lindbladian Reverse Engineering* ($\mathcal{L}$RE), based on both exact methods [22–29] and variational principles [30–39].

On the one hand, $\mathcal{L}$RE based on exact methods allow a precise (exact) reconstruction of the map. However, they are either restricted to a small subset of the possible Lindbaldian evolutions, such as those whose steady states necessarily satisfy a detailed balance condition [22], support pure dark states [23, 24] or are constrained to be Gaussian [25]; or have an impractical computational cost for their implementation, such as in full process tomography [26–28]. In this latter case, given a system with Hilbert space dimension $d$, one must perform $d^2$ measurements in order to reconstruct the corresponding (Krauss operators) map. The cost of the method thus scales exponentially with the dimensionality of the system, and becomes infeasible in most practical cases. Therefore variational methods [30–39] have been proposed in order to fill these gaps, as they are able to reconstruct general maps with reduced computational complexity.

The essential procedure in variational $\mathcal{L}$RE methods is to collect the information about the Lindbladian indirectly by the evolution of a *finite* number of different initial states, observables and/or evolution times. These data do not need to fill all possible measurement outcomes, thus reducing the computational cost but still allowing to reconstruct the Lindbaldian within a controlled accuracy threshold. The reconstruction process follows by searching in the space of possible Lindbladians (the Lindbladian *ansatz*) for the one that best fits the mea-

surement results. The search is usually performed by minimizing a predefined "cost function", such as maximun likelihood estimators for the measurement probability distributions [30, 32, 33, 35], neural network loss functions [38, 39], semidefinite programming [34], among others [31]. It is important to recall that the cost functions are generally either non-linear or non-convex estimators, making the minimization a non-trivial and demanding task.

In this work we propose a variational $\mathcal{L}$RE method for target non-equilibrium steady states (NESS). The method has a significantly reduced complexity on both the required data and the associated Lindbladian estimation cost function. Specifically, given a target NESS, the method requires a number of NESS observables that scales linearly (at most quadratically) with the number of terms in the Lindbladian Hamiltonian (jump operator) ansatz. In addition, the Lindbladian estimator is a linear function of the measurement observables, and the cost function minimization task is mapped to a simple eigenvalue and eigenvector problem. The reconstructed Lindbladian is obtained by simply diagonalizing a *correlation matrix*, where the null eigenvalues correspond to Lindbladian solutions. It is important to note that this is an iff relation, as we discuss, unlike other approaches where the analogous relation is only a necessary one (i.e., the reconstructed Lindbladian is the one that *necessarily* matches to the sampled input data, with no guaranteed extrapolation from it). In this way our method works both as a sufficient condition for the reconstruction of a Lindbladian, as well as a "no-go theorem" whose absence of null eigenvalues in the correlation matrix determines the impossibility of the Lindbladian ansatz to generate the corresponding NESS. In other words, the method has the ability not only to define with certainty, if possible, Lindbaldian evolutions that lead to a desired NESS but also to determine the feasibility of such evolutions in a proposed setup. We note that our proposed method focus only in steady state engineering, i.e., reconstructing a Lindbladian that specifically achieves NESS on its asymptotic long-time dynamics, without discrimination on its finite time properties.

We apply the method in different models, from Gaussian bosonic models to collective spin systems. By systematically exploring the Lindbladian ansatz, we can identify different types of interactions that give rise to the same desired NESS. This knowledge can open interesting perspectives for the field, providing valuable insights into out-of-equilibrium phases and phase transitions, where different phases of matter can emerge from the competition between coherent and dissipative terms of the Lindbladian. Thus, the technique can be used to envision a range of different physical settings capable of generating specific phases of matter, each with great potential for practical applications.

The manuscript is organized as follows. In Sec.II we formulate our method for $\mathcal{L}$RE towards a target nonequilibrium steady state. In Sec.III we illustrate the method

for different systems, bosonic gaussian NESS (III A) and collective spin-1/2 ones (III B). We present our conclusions in Sec.V.

## II. LINDBLADIAN RECONSTRUCTION ALGORITHM

In this paper we consider open quantum system whose dynamics description is given by a Markovian master equation. More specifically, the time evolution for the density matrix $\hat{\rho}$ is described by the the generalized GKS-Lindblad master equation,

$$\frac{d}{dt}\hat{\rho} = \mathcal{L}[\hat{\rho}] = \sum_j^J c_j \mathcal{L}_j^H[\hat{\rho}] + \sum_{j,k}^K \gamma_{j,k} \mathcal{L}_{j,k}^D[\hat{\rho}], \quad (1)$$

where $\mathcal{L}$ is the Lindbladian superoperator, with coherent driving terms

$$\mathcal{L}_j^H = -i[\hat{h}_j, \hat{\rho}], \quad (2)$$

and dissipative ones

$$\mathcal{L}_{j,k}^D = \left( \hat{\ell}_j \hat{\rho} \hat{\ell}_k^\dagger - \frac{1}{2}\{\hat{\ell}_k^\dagger \hat{\ell}_j, \hat{\rho}\} \right). \quad (3)$$

The coherent driving terms are spanned by a set of $J$ Hermitian operators $\{\hat{h}_j\}_{j=1}^J$ with corresponding real coefficients $c_j \in \mathbb{R}$. The dissipative terms are spanned by a set of $K$ jump operators $\{\hat{\ell}_j\}_{j=1}^K$, with corresponding rates $\gamma_{j,k} \in \mathbb{C}$, elements of the dissipative matrix $\boldsymbol{\gamma}$ which is a positive semidefinite matrix, ensuring the complete positivity of the dynamical map. A crucial aspect of $\mathcal{L}$RE is the selection of such a set of Hamiltonian and jump operators. This prior selection is fundamental to the reconstruction process and is inherently related to the physical operations capable of generating the steady state. Thus, the choice of the basis not only reflects the underlying physics of the system but is also consistent with the identification of the physical operations that lead to the non-equilibrium steady states (NESS). In the subsequent sections, we will illustrate how this choice of basis can affect the method.

The reverse engineering approach assumes that the steady state $\hat{\rho}_{ss}$ (i.e., the state reached in the infinite time limit by the dynamics) of the system is known, and asks the question of what are the coefficients $\{c_j\}$ and rates $\{\gamma_{j,k}\}$ of the corresponding Lindbladian necessary to generate such a steady state. In other words, we are interested in solving the steady state equality $\mathcal{L}[\hat{\rho}_{ss}] = 0$ but not from the density matrix perspective, rather from its Lindbladian superoperator. Precisely, we aim at solving the following task:

$$\text{solve} \quad \mathcal{L}[\hat{\rho}_{ss}] = 0 \quad (4)$$

$$\text{with variables} \quad \begin{cases} c_j \in \mathbb{R}, & j = 1, ..., J, \\ \gamma_{j,k} \in \mathbb{C}, & j, k = 1, ..., K. \end{cases} \quad (5)$$

We first notice that given a system with a Hilbert space dimension $d$, the steady state condition of Eq.(4) corresponds to solving a set of $d$ (possibly nonlinear) equalities among the variables of Eq(5). The complexity of this direct approach grows with the Hilbert space dimension, making the solution challenging for general systems (e.g. in many-body 1/2-spin systems whose Hilbert space dimension can grow exponentially with the number $N$ of constituents, $d = 2^N$). Different approaches can be used to avoid dealing with such a highly complex task, as e.g. focusing on specific classes of NESS [22–28] or within variational LRE approaches [30–39]. However, even in variational approaches, the reconstruction task can be reduced to either nonlinear or non-convex estimation problems, which are still nontrival depending on the specific system under study. An approach closer to this work worth remarking, also specifically focused on target NESS, is the one based on solving the Heisenberg equations of motion for specific observables [15, 36]. Although the reconstruction task is linear with the size of the Lindbladian ansatz (similar to ours - as we discuss below), it is not an iff relation and strongly depends on the set of observables chosen to solve within its Heisenberg equations.

In this work we propose a new approach to obtain the corresponding Lindbladian superoperator for a given non-equilibrium steady state. We map the task of Eqs.(4)-(5) into the diagonalization of a $(J+K^2) \times (J+K^2)$ positive semidefinite matrix, $\hat{M}(\hat{\rho}_{\text{ss}})$, thus avoiding the Hilbert space dimensional complexity. In order to do so, we first recall a notion of rapidity for the Lindbladian dynamics, $R(\hat{\rho}) = \text{Tr}(\mathcal{L}[\hat{\rho}]^\dagger \mathcal{L}[\hat{\rho}])$. This function computes the square of the Frobenius norm of the operator $\mathcal{L}[\hat{\rho}]$,

i.e., the squared norm of the time derivative for the state $\hat{\rho}$. On the one hand, if the state is the steady state, the rapidity must vanish. On the other hand, if the rapidity vanishes, since $\mathcal{L}[\hat{\rho}]^\dagger \mathcal{L}[\hat{\rho}]$ is a positive semidefinite operator, the state must be a steady state, $\hat{\rho} = \hat{\rho}_{\text{ss}}$. Therefore, a null rapidity is a necessary and sufficient condition for the Lindbladian steady state,

$$R(\hat{\rho}) = 0 \iff \hat{\rho} = \hat{\rho}_{\text{ss}}. \tag{6}$$

We can reformulate the above relation to simpler terms. We first expand the rapidity using Eq.(1), obtaining that

$$\begin{aligned} R(\hat{\rho}) &= \sum_{j,k} c_j c_k \, \text{Tr}\left(\mathcal{L}_j^H[\hat{\rho}] \mathcal{L}_k^H[\hat{\rho}]\right) \\ &+ \sum_{j,k,m} c_j \gamma_{k,m} \, \text{Tr}\left(\mathcal{L}_j^H[\hat{\rho}] \mathcal{L}_{k,m}^D[\hat{\rho}]\right) \\ &+ \sum_{j,k,m} \gamma_{j,k}^* c_m \, \text{Tr}\left(\mathcal{L}_{j,k}^D[\hat{\rho}]^\dagger \mathcal{L}_m^H[\hat{\rho}]\right) \\ &+ \sum_{j,k,m,n} \gamma_{j,k}^* \gamma_{m,n} \, \text{Tr}\left(\mathcal{L}_{j,k}^D[\hat{\rho}]^\dagger \mathcal{L}_{m,n}^D[\hat{\rho}]\right), \end{aligned} \tag{7}$$

where we use the hermicity of the Lindbladian coherent components, $c_j = c_j^*$ and $\mathcal{L}_j^H[\cdot] = \mathcal{L}_j^H[\cdot]^\dagger$, $\forall j$. The above relation can be written in a matrix form,

$$R(\hat{\rho}) = \langle\varphi_{\mathcal{L}}| \hat{M}(\hat{\rho}) |\varphi_{\mathcal{L}}\rangle, \tag{8}$$

where the $(J+K^2) \times 1$ Lindbladian vector $|\varphi_{\mathcal{L}}\rangle$ concatenates the parameters $c_j$ and $\gamma_{j,k}$ and $\hat{M}(\hat{\rho})$ is a $(J+K^2) \times (J+K^2)$ correlation matrix obtained from the state properties. Specifically,

$$|\varphi_{\mathcal{L}}\rangle = \begin{pmatrix} c_1 \\ \vdots \\ c_J \\ \gamma_{1,1} \\ \gamma_{1,2} \\ \vdots \\ \gamma_{K,K} \end{pmatrix}, \quad \hat{M}(\hat{\rho}) = \begin{pmatrix} \text{Tr}\left(\mathcal{L}_1^H[\hat{\rho}]\mathcal{L}_1^H[\hat{\rho}]\right) & \cdots & \text{Tr}\left(\mathcal{L}_1^H[\hat{\rho}]\mathcal{L}_J^H[\hat{\rho}]\right) & \text{Tr}\left(\mathcal{L}_1^H[\hat{\rho}]\mathcal{L}_{11}^D[\hat{\rho}]\right) & \cdots & \text{Tr}\left(\mathcal{L}_1^H[\hat{\rho}]\mathcal{L}_{K,K}^D[\hat{\rho}]\right) \\ & \ddots & \vdots & \vdots & \ddots & \vdots \\ & & \text{Tr}\left(\mathcal{L}_J^H[\hat{\rho}]\mathcal{L}_J^H[\hat{\rho}]\right) & \text{Tr}\left(\mathcal{L}_J^H[\hat{\rho}]\mathcal{L}_{1,1}^H[\hat{\rho}]\right) & \cdots & \text{Tr}\left(\mathcal{L}_J^H[\hat{\rho}]\mathcal{L}_{K,K}^D[\hat{\rho}]\right) \\ & & & \text{Tr}\left(\mathcal{L}_{1,1}^D[\hat{\rho}]\mathcal{L}_{1,1}^D[\hat{\rho}]\right) & \cdots & \text{Tr}\left(\mathcal{L}_{1,1}^D[\hat{\rho}]\mathcal{L}_{K,K}^D[\hat{\rho}]\right) \\ & \text{H.c.} & & & \ddots & \vdots \\ & & & & & \text{Tr}\left(\mathcal{L}_{K,K}^D[\hat{\rho}]\mathcal{L}_{K,K}^D[\hat{\rho}]\right) \end{pmatrix}. \tag{9}$$

We then notice that, since $\hat{M}(\hat{\rho})$ is a positive semidefinite operator, the rapidity is null *iff* the Lindbladian vector $|\varphi_{\mathcal{L}}\rangle$ is an eigenstate of the correlation matrix with a null eigenvalue. In summary,

$$M(\hat{\rho}) |\varphi_{\mathcal{L}}\rangle = 0 \iff \hat{\rho} = \hat{\rho}_{\text{ss}}. \tag{10}$$

Therefore, we mapped the reverse engineering task Eqs.(4)-(5) to finding the eigenvector with null eigenvalue of the correlation matrix $M(\hat{\rho}_{\text{ss}})$.

A few properties are important remarking:

- The method reduces the reverse engineering complexity to a simpler diagonalization procedure, further reducing the dimensional cost to a $(J+K^2) \times (J+K^2)$ matrix;

- The method gives both a necessary and a sufficient condition for generating the NESS. Thus it not only identifies possible Lindbladians for a given steady state, but could also verify the impossibility to generate a steady state for a given class of Lindbladians, analogous to a "no-go the-

orem". More precisely, given a class of Lindbladians (i.e., using a specific set of Hermitian operators $\hat{h}_j$ and jump operators $\ell_j$ in Eq.(1)) if the correlation matrix $\hat{M}(\hat{\rho}_{\rm ss})$ has no null eigenvalues, one could never reach exactly the corresponding steady steady within this class of dynamics.

- The positive semi-definiteness of the dissipative matrix $\boldsymbol{\gamma}$, a sufficient condition for Markovianity, is not explicitly imposed in the correlation matrix definition. Consequently, solutions that fail to satisfy this condition may not correspond to physical dynamical maps. Checking whether this is the case is usually a difficult task. One could circumvent these issues by post-processing the solution given by the method and explicitly imposing the Markovianity; e.g. once the kernel space of the correlation matrix $\hat{M}(\hat{\rho}_{\rm ss})$ is obtained (possibly degenerate), (i) either a post-selection process is performed to select the solutions satisfying $\boldsymbol{\gamma} \geq 0$, (ii) or given the solution $\boldsymbol{\gamma} \ngeq 0$ one approximates it to a Markovian dissipative matrix. A straightforward approach is to set any negative eigenvalue of the dissipative matrix to zero, especially when these negative eigenvalues are orders of magnitude smaller than the positive ones. We discuss these ideas in more details in the examples of the next section. Recent works have proposed using negative decoherence rates, as they appear in the canonical form of the master equation, to characterize non-Markovianity [42–45]. In this context, it would be interesting to investigate whether the negative eigenvalues of the dissipative matrix obtained within our framework could be interpreted as signatures of non-Markovian dynamics, associated with system–environment interactions that allow partial reversals of earlier dissipative processes. We leave such an analysis as a perspective for future work.

Before presenting our results for specific examples, we outline below the general procedure used to apply the method.

*Step 1.* Select a target NESS and define suitable operator bases for both the Hamiltonian ($\{\hat{h}_j\}$) and the jump operators ($\{\hat{\ell}_j\}$), chosen to capture the relevant physical configurations of interest.

*Step 2.* Construct the corresponding correlation matrix (Eq.(9)) and determine its kernel. If the kernel is empty, one may return to Step 1 and enlarge the operator bases. Alternatively, the dynamics associated with the smallest eigenvalue of the correlation matrix can be analyzed.

*Step 3.* Once a solution ($|\varphi_L\rangle$) is obtained, its physicality must be verified. This involves checking that, up to a global phase, the Hamiltonian coefficients are real and that the dissipative matrix is positive semidefinite. If these conditions are not met, one may either return to Step 1 or approximate the solution by a Markovian dynamical model, as detailed above.

We illustrate this procedure in the following section through explicit examples.

## III. EXAMPLES

In this section we apply our method to different systems, namely, with (i) bosonic Gaussian steady states, (ii) spin systems with collective dissipation and (iii) spin models with random short-range interactions and local dissipation. These quantum states possess unique properties that can play a pivotal role in the development of quantum technologies [46], therefore with great interest for engineering methods. Moreover, these are well-established and extensively studied systems with analytical results that help to illustrate important aspects of the method.

### A. Bosonic Gaussian States

We first consider the reverse engineering of Lindbladians generating single-mode bosonic Gaussian steady states, as coherent or squeezed vacuum states.

#### 1. Coherent steady states

Coherent states hold significant importance within the realm of quantum physics, especially in the domain of quantum optics [47]. These are states of the quantum harmonic oscillator with minimal uncertainty (minimum quantum noise in the canonical conjugate variables, specifically the quadratures) and exhibit the most analogous evolution to the classical harmonic oscillator [48], i.e. $\Delta X \Delta Y = \frac{1}{4}$, with $\Delta X = \Delta P = \frac{1}{2}$, where $\Delta A = \sqrt{\langle \hat{A}^2 \rangle - \langle \hat{A} \rangle^2}$, quadratures $\hat{X} = \frac{1}{\sqrt{2}}\left(\hat{a}^\dagger + \hat{a}\right)$, $\hat{P} = \frac{i}{\sqrt{2}}\left(\hat{a}^\dagger - \hat{a}\right)$, and $\hat{a}(\hat{a}^\dagger)$ is the bosonic annihilation (creation) operator. A single-mode coherent state is described as [47, 48],

$$|\alpha\rangle \equiv \hat{D}(\alpha)|0\rangle = \exp\left(\alpha\hat{a}^\dagger - \alpha^*\hat{a}\right)|0\rangle, \qquad (11)$$

where $|0\rangle$ is the ground state of the harmonic oscillator, i.e. $\hat{a}|0\rangle = 0$, and $\hat{D}(\alpha)$ is known as the displacement operator, $\alpha \in \mathbb{C}$. The coherent state is an eigenstate of the annihilation operator with eigenvalue $\alpha$, i.e. $\hat{a}|\alpha\rangle = \alpha|\alpha\rangle$ and the displacement operator is an unitary that shifts the annihilation operator by $\alpha$, $\hat{D}(\alpha)^\dagger \hat{a}\hat{D}(\alpha) = \hat{a} + \alpha$.

We assume single particle bosonic operators for the coherent and dissipative operator basis in the Lindbladian (Eq.(1)): $\{\hat{h}_j\} = \{(\hat{a} + \hat{a}^\dagger)/\sqrt{2}, (\hat{a} - \hat{a}^\dagger)/i\sqrt{2}\}$ and $\{\hat{\ell}_j\} = \{\hat{a}, \hat{a}^\dagger\}$. This choice is reasonable since coherent states can be generated with the displacement operator,

a function of linear operators, acting in the vacuum. The correlation matrix can be written as,

$$\hat{M}(|\alpha\rangle\langle\alpha|) = \begin{pmatrix} 1 & 0 & -\frac{\sqrt{2}}{4}i(\alpha-\alpha^*) & 0 & 0 & \frac{\sqrt{2}}{4}i(\alpha-\alpha^*) \\ & 1 & -\frac{\sqrt{2}}{4}(\alpha+\alpha^*) & 0 & 0 & \frac{\sqrt{2}}{4}(\alpha+\alpha^*) \\ & & \frac{1}{2}|\alpha|^2 & 0 & 0 & -\frac{1}{2}|\alpha|^2 \\ & & & \frac{1}{2} & 0 & 0 \\ & \text{H.c.} & & & \frac{1}{2} & 0 \\ & & & & & 2+\frac{1}{2}|\alpha|^2 \end{pmatrix}. \tag{12}$$

The kernel of $\hat{M}(|\alpha\rangle\langle\alpha|)$ is one-dimensional. Thus, in this scenario the method provides a unique eigenvector with null eigenvalue,

$$|\varphi_{\mathcal{L}}\rangle_\alpha = \begin{pmatrix} -\frac{\sqrt{2}}{4i}(\alpha-\alpha^*) \\ \frac{\sqrt{2}}{4}(\alpha+\alpha^*) \\ 1 \\ 0 \\ 0 \\ 0 \end{pmatrix}, \tag{13}$$

or in other words, a unique Lindbladian spanned in the basis $\{h_j\}$ and $\{\ell_j\}$ leading to such coherent steady states, characterized by the master equation,

$$\frac{d}{dt}\hat{\rho} = -i[\hat{H},\hat{\rho}] + \left(\hat{L}\hat{\rho}\hat{L}^\dagger - \frac{1}{2}\{\hat{L}^\dagger\hat{L},\hat{\rho}\}\right), \tag{14}$$

with,

$$\hat{H} = -\frac{i}{2}\alpha^*\hat{a} + \frac{i}{2}\alpha\hat{a}^\dagger, \quad \hat{L} = \hat{a}. \tag{15}$$

The coherent state emerges from a non-trivial interplay between the unitary and dissipative parts of the model, i.e. $[\hat{H},\hat{L}] \neq 0$. While attaining this specific solution might not pose significant challenges in this preliminary illustration, its significance also lies in demonstrating that, under the assumption of linear Hamiltonian and linear jump operators—specifically, $\{\hat{h}_j\} = \{(\hat{a}+\hat{a}^\dagger)/\sqrt{2},$

$(\hat{a}-\hat{a}^\dagger)/i\sqrt{2}\}$ and $\{\hat{\ell}_j\} = \{\hat{a},\hat{a}^\dagger\}$—the method yields a unique solution in Lindblad form, $|\varphi_{\mathcal{L}}\rangle_\alpha$, up to a multiplicative factor, within the chosen basis.

### 2. Squeezed vacuum steady states

Squeezed vacuum are states of minimum uncertainty but the noise in one of the quadratures is below of corresponding noise in the vacuum state, consequently the noise of the other quadrature is amplified [47, 48], i.e. $\Delta X \Delta Y = \frac{1}{4}$, with $\Delta X = \frac{1}{2}e^{-r}$ and $\Delta P = \frac{1}{2}e^r$, where $r$ is called the squeezed parameter. Such states play an important role in quantum metrology, as e.g. improving laser interferometers [49, 50]. Moreover, they hold great potential for applications in the field of quantum cryptography, fortifying secure optical communication [51, 52]. A single-mode squeezed vacuum state can be defined as,

$$|\xi\rangle \equiv \hat{S}(\xi)|0\rangle = \exp\left(\xi^*\hat{a}^2 - \xi(\hat{a}^\dagger)^2\right)|0\rangle, \tag{16}$$

where $\hat{S}(\xi)$ is the squeezed operator and $\xi = re^{i\theta}$, with $r > 0$ and $\theta \in \mathbb{R}$. The squeezed vacuum state is an eigenstate of the operator $\hat{b} = \hat{a}\cosh(r) + \hat{a}^\dagger e^{i\theta}\sinh(r)$ with eigenvalue zero, i.e. $\hat{b}|\xi\rangle = 0$. The squeezed operator is an unitary acting on the annihilation operator $\hat{a}$ as a Bogoliubov transformation, $\hat{S}(\xi)^\dagger\hat{a}\hat{S}(\xi) = \hat{a}\cosh(r) - \hat{a}^\dagger e^{i\theta}\sinh(r)$.

Squeezed states can be generated through non-linear processes resulting from the interaction between bosons, involving quadratic Hamitonians [53, 54]. We therefore apply our method expanding the Hamiltonian basis with bosonic quadratic operators, $\{\hat{h}_j\} = \{(\hat{a}^2+(\hat{a}^\dagger)^2)/\sqrt{2},$ $(\hat{a}^2-(\hat{a}^\dagger)^2)/i\sqrt{2}\}$. To demonstrate the applicability of the method in identifying different dynamics that produces the target states, we will select two jump operators basis.

*Single particle jump operators.-* As a first attempt, we propose jump operators spanned by single particle operators: $\{\hat{\ell}_j\} = \{\hat{a},\hat{a}^\dagger\}$. Note that from the relation $\hat{b}|\xi\rangle = 0$ we can already infer one possible solution, a purely dissipative dynamics, $\hat{H} = 0$, with a single lindblad jump operator by $\hat{\ell} \equiv \hat{b}$. In fact, the correlation matrix $\hat{M}(|\xi\rangle\langle\xi|)$ (see Appendix (A)) has a three-dimensional kernel and can be expanded by the orthogonal vectors,

$$|\varphi_{\mathcal{L}_1}\rangle_\xi = \begin{pmatrix} -\frac{\sqrt{2}}{2}\sin(\theta) \\ \frac{\sqrt{2}}{2}\cos(\theta) \\ -\tanh(2r) \\ e^{-i\theta}\text{sech}(2r) \\ e^{i\theta}\text{sech}(2r) \\ \tanh(2r) \end{pmatrix}, \quad |\varphi_{\mathcal{L}_2}\rangle_\xi = \begin{pmatrix} \frac{\sqrt{2}}{2}\cos(\theta) \\ \frac{\sqrt{2}}{2}\sin(\theta) \\ 0 \\ ie^{-i\theta}\cosh(2r) \\ -ie^{i\theta}\cosh(2r) \\ 0 \end{pmatrix} \quad \text{and} \quad |\varphi_{\mathcal{L}_3}\rangle_\xi = \frac{1}{2}\begin{pmatrix} 0 \\ 0 \\ \cosh(2r)+1 \\ e^{-i\theta}\sinh(2r) \\ e^{i\theta}\sinh(2r) \\ \cosh(2r)-1 \end{pmatrix}. \tag{17}$$

The first two solutions have non-positive dissipative matrix, $\{\lambda_j(\gamma_1)\} = \{-1,1\}$ and $\{\lambda_j(\gamma_2)\} =$

$\{-\cosh(2r),\cosh(2r)\}$, where $\lambda_j(\boldsymbol{A})$ is the $j$-th eigenvalue of $\boldsymbol{A}$ and $\gamma_k$ is the dissipative matrix of the $k$-

th solution. Therefore, we cannot guarantee that they represent physical dynamics. The third solution is the previously commented purely dissipative dynamics, characterized by $\hat{H} = \hat{0}$ and $\hat{\ell} = \hat{b}$.

We can explore alternative Markovian solutions by examining the superposition of the three eigenstates of $\hat{M}(|\xi\rangle\langle\xi|)$. Assuming $|\varphi_{\mathcal{L}}\rangle_\xi = \sum_{j=1}^3 a_j |\varphi_{\mathcal{L}_j}\rangle_\xi$, with $a_j \in \mathbb{R}\ \forall j$, the dissipative matrix is positive semi-definite, i.e. $\gamma \geq \mathbf{0}$ iff $a_1 = a_2 = 0$. Therefore, for the basis choice, linear jump operators, the only Markovian dynamics will be governed by the purely dissipative solution $|\varphi_{\mathcal{L}_3}\rangle_\xi$. The possibility of obtaining a Markovian solution, where the squeezed vacuum steady state emerges through the interaction between the unitary and dissipative components, is explored in the subsequent discussion by considering an alternative basis for the dissipative term.

*Two-particle jump operators.-* Now, considering an interaction among the bosons arising from their coupling with the environment, we expand the jump operators by two-particle operators: $\{\hat{\ell}_j\} = \{\hat{a}^2, (\hat{a}^\dagger)^2\}$. We compute the correlation matrix $\hat{M}(|\xi\rangle\langle\xi|)$ (see Appendix (A)) which has an one-dimensional kernel,

$$|\varphi_{\mathcal{L}}\rangle_\xi = \begin{pmatrix} \sqrt{2}\sin(\theta)\sinh(4r) \\ -\sqrt{2}\cos(\theta)\sinh(4r) \\ 3 + 4\cosh(2r) + \cosh(4r) \\ e^{-2i\theta}(1 - \cosh(4r)) \\ e^{2i\theta}(1 - \cosh(4r)) \\ 3 - 4\cosh(2r) + \cosh(4r) \end{pmatrix}. \quad (18)$$

Therefore, in this scenario the dynamics can be described by the master equation,

$$\hat{H} = i\left(e^{-i\theta}\hat{a}^2 - e^{i\theta}\left(\hat{a}^\dagger\right)^2\right)\sinh(4r),$$

$$\hat{\ell} = -e^{-2i\theta}\sqrt{2}(\cosh(2r)+1)\hat{a}^2 + \sqrt{2}(\cosh(2r)-1)\left(\hat{a}^\dagger\right)^2. \quad (19)$$

Interestingly, we observe in this way that by postulating interactions between bosons resulting from their coupling with the environment, the competition between Hamiltonian and dissipative dynamics can manifest as a squeezed state in long time regime.

## B. Driven-dissipative Collective Spin Model

We study in this section the method for a collective spin system. The model describes a set of $N$ 1/2-spin systems collectively coupled to a Markovian environment, leading to a GKS-Lindbladian master equation evolution

$$\frac{d}{dt}\hat{\rho} = \mathcal{L}_{DD}[\hat{\rho}] \equiv -i\omega_o[\hat{S}^x, \hat{\rho}] + \frac{\kappa}{S}\left(\hat{S}_-\hat{\rho}\hat{S}_+ - \frac{1}{2}\{\hat{S}_+\hat{S}_-, \hat{\rho}\}\right), \quad (20)$$

where $S = N/2$ is the total spin of the system, $\hat{S}^\alpha = \sum_j \hat{\sigma}_j^\alpha/2$ with $\alpha = x, y, z$ are collective spin operators,

$\hat{S}_\pm = \hat{S}^x \pm iS^y$ are the excitation and decay operators, and $\hat{\sigma}_j^\alpha$ is the Pauli spin operator for the $j$'th spin. Inheriting the $SU(2)$ algebra of their constituents, the collective operators satisfy the commutation relations $[\hat{S}^\alpha, \hat{S}^\beta] = i\epsilon^{\alpha\beta\gamma S^\gamma}\hat{S}^\gamma$. Due to the collective nature of their interactions, the model conserves the total spin $S^2 = (\hat{S}^x)^2 + (\hat{S}^y)^2 + (\hat{S}^z)^2$. The model encompasses an interplay between coherent driving and incoherent decay, with coherent rate $\omega_o$ and an effective decay rate $\kappa$. It is commonly used to describe cooperative emission in cavities [55–58] and was recently shown to support a time crystal phase with the spontaneously breaking of continuous time-translational symmetry [59, 60]. In the strong dissipative regime, $\kappa/\omega_0 > 1$, the spins in the steady state predominantly align in the spin-down direction along $z$-axis. Conversely, in the weak dissipative case, $\kappa/\omega_0 < 1$, the dynamics is characterized by persistent temporal oscillations of macroscopic observables [59] and a continuous growth of correlations [61, 62].

The steady states of the Lindbladian are obtained analytically [57, 58], with the form

$$\hat{\rho}_{\rm ss} = \frac{1}{\mathcal{N}}\hat{\eta}^\dagger\hat{\eta} \quad \text{with} \quad \hat{\eta} = \sum_{j=0}^{N+1}\left(\frac{\hat{S}_-}{-i\omega_o N/2\kappa}\right)^j, \quad (21)$$

where $\mathcal{N}$ is the normalization constant. Due to the collective nature of the steady state we chose the basis for the Lindbladian also composed of collective operators $\{\hat{h}_j\} = \{\hat{S}_x, \hat{S}_y, \hat{S}_z\}$ and $\{\hat{\ell}_j\} = \{\hat{S}_x, \hat{S}_y, \hat{S}_z\}$. Employing the method numerically for the above steady states in systems with finite size $N$ we recover the exact Lindbladian dynamics of Eq.(20). Precisely, we obtain a unique eigenvector $|\varphi_{\mathcal{L}}\rangle$ in the kernel of the correlation matrix $\hat{M}(\hat{\rho}_{\rm ss})$, with elements given by,

$$c_1 = \omega_0, \quad c_2 = c_3 = 0,$$
$$\gamma_{1,1} = \gamma_{2,2} = \sqrt{\kappa}, \quad \gamma_{2,1} = \gamma_{1,2}^* = i\sqrt{\kappa}, \quad (22)$$
$$\gamma_{3,1} = \gamma_{1,3}^* = \gamma_{3,2} = \gamma_{2,3}^* = \gamma_{3,3} = 0.$$

We illustrate the behavior in the weak dissipative case, $\omega_0/\kappa = 2$ . In Fig.(1a) we show the absolute value of the obtained Lindbladian parameters, for different system sizes. We observe the nullity (i.e., below the numerical accuracy of the order $\approx O(10^{-12})$) of parameters $c_2$, $c_3$ and $\gamma_{3,j}$ with $j = (1, 2, 3)$, in addition to the information that $|\gamma_{1,k}| = |\gamma_{2,k}|$ for $k = (1, 2)$.

The two smallest eigenvalues of the correlation matrix are displayed in Fig.(1b). For finite sizes we observe a unique kernel solution (i.e., with eigenvalue below the numerical accuracy of the order $\approx O(10^{-12})$), while observing a decrease in the second eigenvalue for larger $N$. It is well-known that the model is gapless in the weak dissipative regime [59, 63]. The vanishing of the second eigenvalue with the increasing system size $N$ (inset panel of Fig.(1b)) appears to capture this characteristic of the model. The eigenvalues of the dissipative matrix $\boldsymbol{\gamma}$ for the solution (kernel) of $\hat{M}(\hat{\rho}_{\rm ss})$ are showed in Fig.(1c).

Observe that the dissipative matrix for the kernel solution has just one nonnull eigenvalue, confirming that we can associate it with only one jump operator (collective decay).

### C. Random Local Model

We now turn to a many-body local model of a spin-1/2 chain with random local interactions and single-site jump operators [5]. The model Hamiltonian includes both on-site and nearest-neighbor two-body interaction terms,

$$\hat{H}_j = \sum_{j,\alpha} c_{j,\alpha}\hat{\sigma}_j^\alpha + \sum_{j,\alpha,\beta} c_{j,\alpha,\beta}\hat{\sigma}_j^\alpha \hat{\sigma}_{j+1}^\beta, \qquad (23)$$

while the dissipative dynamics is driven by local operators of the form

$$\hat{L}_j = \sum_\alpha d_{j,\alpha}\hat{\sigma}_j^\alpha, \qquad (24)$$

where $\hat{\sigma}_j^\alpha$ indicates the Pauli operator in the $\alpha \in \{x, y, z\}$ direction acting on the $j$'th site. The Hamiltonian coefficients $c_{j,\alpha}$ and $c_{j,\alpha,\beta}$ are sampled from a Gaussian distribution with zero mean and unit variance, under open boundary conditions $c_{N,\alpha,\beta} = 0$. Similarly, the real and imaginary parts of the dissipative coefficients $d_{j,\alpha}$ are drawn from independent Gaussian distributions with zero mean and standard deviation $\alpha_D$, which sets the overall strength of the dissipation.

We analyze chains of size $N = 5$ over the range $\alpha_D \in (0, \sqrt{10})$. For each random realization of the coefficients, the non-equilibrium steady state (NESS) is obtained via exact diagonalization of the Lindbladian superoperator. In applying the procedure described in Sec. II, we adopt an ansatz basis consisting of single and two-body Pauli operators for the Hamiltonian terms, $\{\hat{h}_{j,k}^{\alpha,\beta}\} = \{\sigma_j^\alpha, \sigma_j^\alpha \sigma_k^\beta\}_{j,k=1}^N$, and single-body Pauli operators for the jump operators $\{\hat{\ell}_j^\alpha\} = \{\sigma_j^\alpha\}_{j=1}^N$, with $\alpha, \beta = x, y, z$.

The two smallest eigenvalues of the correlation matrix are shown in Fig.(2a). In this regime, the kernel of $\hat{M}$ is one-dimensional (values below $\approx O(10^{-12})$ are assumed null within numerical accuracy), uniquely determining the reconstructed Lindbladian parameters consistent with the NESS for $\alpha_D \lesssim 1$. On the other hand, for large dissipative strength $\alpha_D \gtrsim 1$, we observe the closing of the eigenvalue gap, therefore indicating that due to the emerging degeneracy the method may not be fully precise in this regime. Indeed, in order to assess the accuracy of the method, we compute the steady state obtained from the reconstructed Lindbladian solution, $\hat{\rho}_s^M$, and compare it with the expected exact NESS, $\hat{\rho}_s$, obtained from solving the Lindbladian of Eqs. (23) and (24). The results are displayed in Fig.(2b), where their norm difference remains close to zero throughout the $\alpha_D \lesssim 1$ regime, confirming the reliability of the reconstruction in the weak dissipative regime, while becoming uncertain in the strong case, due to the emerging degeneracy.

## IV. ROBUSTNESS OF THE METHOD

In this section, we investigate the robustness of the method against random fluctuations in the target steady state. Specifically, we consider a target steady state represented in the following form

$$\hat{\rho}_\epsilon = \frac{1}{\mathcal{N}_\epsilon}\left[\epsilon\hat{\mathbb{I}}/\text{tr}\left(\hat{\mathbb{I}}\right) + (1 - \epsilon)\hat{\rho}_s\right], \qquad (25)$$

with $\epsilon$ the strength of the random fluctuations mixing the unperturbed steady state with white noise represented by the identity matrix $\hat{\mathbb{I}}$. The state $\hat{\rho}_s$ denotes the NESS of the model, and $\mathcal{N}_\epsilon$ is the normalization constant.

### A. Driven-dissipative Collective Spin Model

We first apply our method to the perturbed steady state $\hat{\rho}_s^\epsilon$ of the collective spin model, using $\hat{\rho}_s$ as given in Eq.(21). We expand the Hamiltonian and jump ansatz operators by collective operators along the $x$ and $y$ directions, $\{\hat{h}_j\} = \{\hat{S}_x, \hat{S}_y\}$ and $\{\hat{\ell}_j\} = \{\hat{S}_x, \hat{S}_y\}$.

*Strong dissipative case $(\kappa/\omega_0 > 1)$.* We first observe that for any $\epsilon \neq 0$ perturbation, the correlation matrix $\hat{M}(\hat{\rho}_\epsilon)$ has no null eigenvalues - see Fig.(3a). Specifically, we find that the smallest eigenvalue has a quadratic dependence with the perturbation strength and decays with system size,

$$\lambda_1(\hat{M}(\hat{\rho}_\epsilon)) \sim \epsilon^2/N, \qquad (26)$$

with $\epsilon \ll 1$. The nonull eigenvalues for the correlation matrix do not guarantee the direct application of our method, i.e., that $\hat{\rho}_\epsilon$ is indeed a steady state of the reverse engineered Lindbladian. On the other hand, due to the vaninshing of the smallest eigenvalue with system size it suggests that the method should still be feasible for large system sizes. We therefore investigate the reversed engineered Lindbladian $\mathcal{L}_\epsilon$ corresponding to this minimum eigenvalue. We also remark here that the eigenvalues for the dissipative matrix $\lambda(\gamma)$ of $\mathcal{L}_\epsilon$ are all nonnegative, therefore assuring the complete positivity for the map.

In Fig.(3b) we show how the steady state of $\mathcal{L}_\epsilon$, denoted by $\hat{\rho}_s^\epsilon$, compares to the unperturbed one $\hat{\rho}_s$ through their norm difference. We obtain that,

$$\|\hat{\rho}_s^\epsilon - \hat{\rho}_s\| \sim \epsilon/N^{\frac{3}{2}}. \qquad (27)$$

Hence we see that the Lindbaldian $\mathcal{L}_\epsilon$ can still be used to generate a steady state similar to the exact (unperturbed) one, with the level of precision increasing as one increases the system size.

*Weak dissipative case $(\kappa/\omega_0 < 1)$.* This case has subtle properties which require a more careful analysis. Similar to the strong dissipative case the correlations matrix

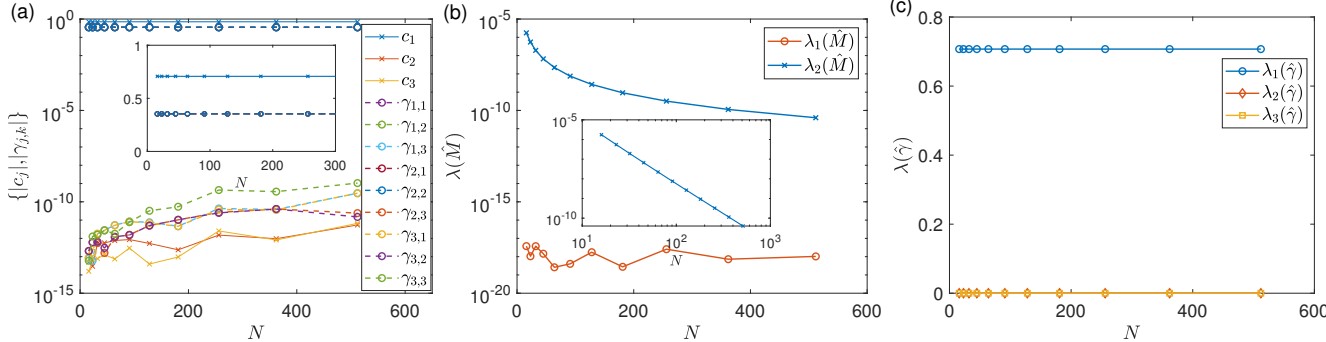

Figure 1. Lindbladian reverse engineering for the collective steady states of Eq.(21), in the weak dissipative regime $\omega/\kappa = 2$. In all panels values below $\approx O(10^{-10})$ are beyond our numerical accuracy, and are interpreted as effectively null. We show in **(a)** the elements of the kernel eigenvector $|\varphi_{\mathcal{L}}\rangle$ for the correlation matrix $\hat{M}(\hat{\rho}_{ss})$. The elements correspond to the set of parameters in the constructed Lindbladian. In panel **(b)** we show the two lowest eigenvalues of the correlation matrix $\hat{M}(\hat{\rho}_{ss})$, displaying in the (inset-panel) only the second eigenvalue in a log-log scale, in order to highlight its gapless behavior. In panel **(c)** we show the eigenvalues of the dissipative matrix $\gamma$ for the lowest eigenstate (kernel) of the correlation matrix.

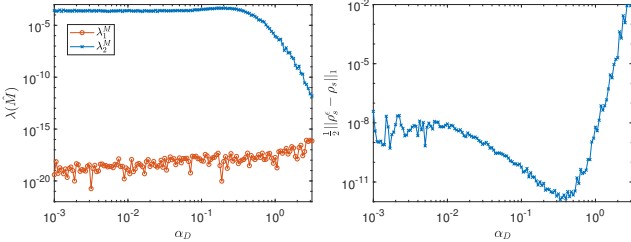

Figure 2. **(a)** The two lowest eigenvalues of the correlation matrix $\hat{M}(\hat{\rho}_s)$ in the random local model, for varying dissipative strength $\alpha_D$. **(b)** The norm difference between the steady obtained from the reverse engineered Lindbladian $\hat{\rho}_s^M$ and the exact one $\hat{\rho}_s$. All results here are averaged over 200 random realizations of the system coefficients.

$\hat{M}(\hat{\rho}_\epsilon)$ has no null eigenvalues given an $\epsilon \neq 0$ perturbation, with the smallest eigenvalue following a scaling relation similar to Eq.(26) - see Fig.(3c). Therefore we could once again consider the reversed engineered Lindbladian $\mathcal{L}_\epsilon$ related to this minimun eigenvalue. However, the dissipative matrix $\hat{\gamma}$ of such Lindbaldian is not positive; specifically, it has one negative eigenvalue. As we discuss, despite being orders of magnitude smaller than the positive eigenvalues, the presence of this negative eigenvalue precludes the assurance of complete positivity for the map. One approach to addres this issue is to disregard the negative eigenvalue of the dissipative matrix by making it null, i.e., with the transformation

$$\lambda_j(\hat{\gamma}) = \max\left(\lambda_j(\hat{\gamma}), 0\right), \quad \forall j. \qquad (28)$$

In this way the corresponding Lindbaldian, which we denote by $\mathcal{L}_\epsilon^+$, is now a complete positivity map by definition.

In Fig.(3d) we show how the steady state of $\mathcal{L}_\epsilon$ and $\mathcal{L}_\epsilon$, denoted as $\hat{\rho}_\epsilon$ and $\hat{\rho}_s^{\epsilon,+}$, respectively, compare to the unpertubed one $\hat{\rho}_s$. The results in both cases show similar behaviors. The curves exhibit a highly non-linear

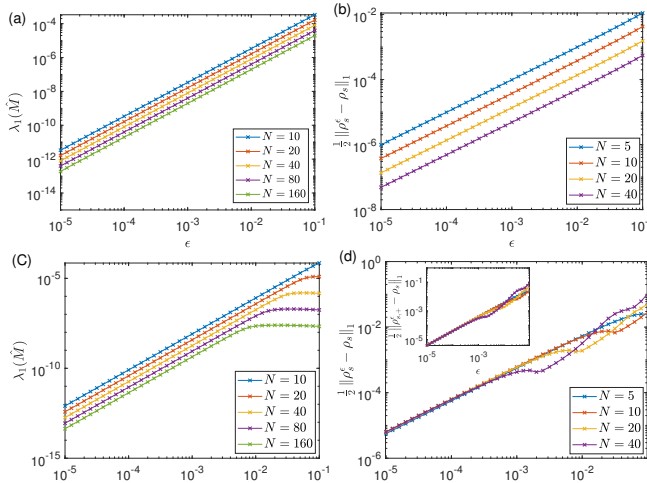

Figure 3. **(a)** The smallest eigenvalue $\lambda_1(\hat{M}_\epsilon)$ of the correlation matrix $\hat{M}(\hat{\rho}_\epsilon)$ as a function of the perturbation $\epsilon$, for different sizes $N$ and in the strong dissipative regime, $\omega_0/\kappa = 1/2$. **(b)** We show the norm difference between $\hat{\rho}_s^\epsilon$ and $\hat{\rho}_s$ in the strong dissipative regime. **(c)** The first eigenvalue of the correlation matrix at the weak dissipative regime, $\omega_0/\kappa = 2$. **(d)** The norm difference between $\hat{\rho}_s^\epsilon$ and $\hat{\rho}_s$ and (inset panel) between $\hat{\rho}_s^{\epsilon,+}$ and $\hat{\rho}_s$, for $\omega_0/\kappa = 2$.

behavior for perturbations greater than an $\epsilon_{\text{saturation}}$ (note the logarithmic scale), which gradually transition to a smoother, linear behavior with a scaling similar to Eq.(27) for smaller perturbations. The value of $\epsilon_{\text{saturation}}$ correlates with the size $N$, specifically, $\epsilon_{\text{saturation}}$ decreases as $N$ increases. It is well-established that the decay time $\tau$ for the collective spin model exhibits an exponential increase with size [59, 63]; particularly, in the thermodynamic limit the decay rate vanishes and the lifetime of oscillations diverge. The numerical simulations indicate that $\tau \sim 1/\epsilon_{sat}$, thus, for long lifetime dynamics, the perturbative term exerts a greater influence.

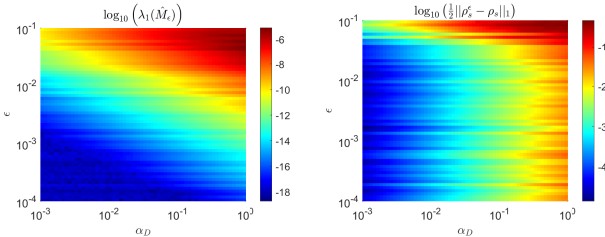

Figure 4. **(a)** The smallest eigenvalue $\lambda_1(\hat{M}_\epsilon)$ of the correlation matrix $\hat{M}(\hat{\rho}_\epsilon)$, for varying perturbation $\epsilon$ and dissipative strength $\alpha_D$. **(b)** The norm difference between the steady obtained from the reverse engineered Lindbladian, $\hat{\rho}_s^\epsilon$, and the exact one $\hat{\rho}_s$. All figures are shown on a logarithmic scale. The results presented here are averaged over 30 random realizations of the system coefficients.

## B.   Random Local Model

We now extend the robustness analysis to the random local model of Sec. III C, applying the method to the perturbed steady state defined in Eq. (25), with $\hat{\rho}_s$ obtained from numerically solving the NESS of Eqs. (23) and (24).

We first observe that, for certain ranges of $\alpha_D$ and $\epsilon$, the correlation matrix $\hat{M}_\epsilon$ exhibits no null eigenvalues, as shown in Fig. (4a). Therefore, following the same procedure as in the previous section, we consider the reverse-engineered Lindbladian $\mathcal{L}_\epsilon$ associated with the smallest eigenvalue. The Lindbladian $\mathcal{L}_\epsilon$ may exhibit a non–positive dissipative matrix, $\hat{\gamma}_\epsilon$. Complete positivity of the dynamics is then enforced using the prescription of Eq. (28). In Fig.(4b) we show how the steady state of $\mathcal{L}_\epsilon^+$, denoted $\hat{\rho}_s^\epsilon$, compares with the unperturbed one $\hat{\rho}_s$ through the logarithm of their norm difference. The Lindbladian $\mathcal{L}_\epsilon$ is capable of reproducing a non-equilibrium steady state closely matching the exact one, with improved accuracies at smaller perturbations $\epsilon$ and dissipative strength $\alpha_D$.

## V.   CONCLUSIONS AND PERSPECTIVES

We have introduced a method for determining Lindbladian superoperators from a non-equilibrium steady state. The method is based on the identification of the kernel of a correlation matrix obtained from the NESS and the definition of a basis for the Lindbladian. The existence of a null element in the domain kernel together with the positive semidefiniteness of the dissipative matrix gives an iff condition for reconstruction of the Lidbladian that generates the NESS. By exploring the method in different systems, we also observed how different Lindbladian bases can associate different types of dynamics with the same NESS. In a future work, it would be interesting to use this property to study dissipative phases and phase transitions. Furthermore, it would be interesting to investigate the elements of the correlation matrix kernel that, despite exhibiting negative decoherence rates, could describe the underlying non-Markovian dynamics of the physical systems

Another promising applications of our method is that of finding alternative Lindbladians for a given target NESS. Specifically, given a NESS generated by a specific Lindbladian $\mathcal{L}_{\text{full}}$, one may address the question if this same state could be engineered by an alternative Lindbladian $\mathcal{L}_{\text{alt}} \neq \mathcal{L}_{\text{full}}$. The alternative Lindbladian could potentially be more experimentally feasible, bringing the preparation and use of the NESS closer to the realms of practicality. Our preliminary studies are encouraging. Analysing the random spin model and its corresponding input NESS ($\hat{\rho}_{exact,NESS}$) under such perspective, we used our method to reverse-engineer a Lindbladian under a constrained ansatz: we forbid dissipative channels at certain sites but allowed for longer-range Hamiltonian interactions. The method successfully found an alternative Lindbladian whose steady state had a fidelity of $\approx 99\%$ with the original target state ($\hat{\rho}_{exact,NESS}$). The obtained reverse engineered Lindbladian dealt with the lack of dissipation by exploiting the Hamiltonian interactions in a longer-range, thereby recovering the same class of NESS. This suggests that a desired complex non-equilibrium steady state (NESS) could potentially be engineered by a different, and possibly more experimentally feasible, set of controls — for example, trading challenging local dissipation for tunable long-range interactions.

## VI.   ACKNOWLEDGEMENTS

We acknowledge financial support from the Brazilian funding agencies CAPES, CNPQ and FAPERJ (Grant No. 308205/2019-7, No. E-26/211.318/2019, No. 151064/2022-9 and E-26/201.365/2022) and by the Serrapilheira Institute (Grant No. Serra 2211-42166).

## Appendix A: Correlation Matrices for Squeezed Vacuum NESS

In this Appendix we show correlation matrices $\hat{M}_\xi$ obtained from choosing the squeezed vacuum as steady state, $\hat{\rho}_s = |\xi\rangle\langle\xi|$, and quadratic operators for the Hamiltonian basis, $\{\hat{h}_j\} = \{(\hat{a}^2 + (\hat{a}^\dagger)^2)/\sqrt{2}, (\hat{a}^2 - (\hat{a}^\dagger)^2)/i\sqrt{2}\}$.

### 1.   Single particle jump operators

For one body jump operators $\{\hat{\ell}_j\} = \{\hat{a}, \hat{a}^\dagger\}$, the correlation matrix is given by

$$M_{11}^{\xi} = \tfrac{1}{2}\left(3 - \cos(2\theta) + (1 + \cos(2\theta))\cosh(4r)\right), \quad M_{12}^{\xi} = \tfrac{1}{2}\sin(2\theta)(\cosh(4r) - 1),$$

$$M_{13}^{\xi} = -\tfrac{\sqrt{2}}{2}\sin(\theta)\sinh(2r), \quad M_{14}^{\xi} = M_{15}^{\xi*} = \tfrac{i\sqrt{2}}{2}\cosh(2r), \quad M_{16}^{\xi} = -\tfrac{\sqrt{2}}{2}\sin(\theta)\sinh(2r),$$

$$M_{22}^{\xi} = \tfrac{1}{2}\left(3 + \cos(2\theta) + (1 - \cos(2\theta))\cosh(4r)\right), \quad M_{23}^{\xi} = \tfrac{\sqrt{2}}{2}\cos(\theta)\sinh(2r),$$

$$M_{24}^{\xi} = M_{25}^{\xi*} = -\tfrac{\sqrt{2}}{2}\cosh(2r), \quad M_{26}^{\xi} = \tfrac{\sqrt{2}}{2}\cos(\theta)\sinh(2r), \tag{A1}$$

$$M_{33}^{\xi} = \tfrac{5}{8} - \cosh(2r) + \tfrac{3}{8}\cosh(4r), \quad M_{34}^{\xi} = M_{35}^{\xi*} = \tfrac{1}{2}e^{i\theta}(\sinh(2r) - \tfrac{3}{4}\sinh(4r)),$$

$$M_{36}^{\xi} = \tfrac{3}{8}(\cosh(4r) - 1), \quad M_{44}^{\xi} = M_{55}^{\xi} = \tfrac{1}{8}(1 + 3\cosh(4r)), \quad M_{45}^{\xi} = \tfrac{3}{8}e^{-2i\theta}(\cosh(4r) - 1),$$

$$M_{46}^{\xi} = M_{56}^{\xi*} = -\tfrac{1}{2}e^{-i\theta}(\sinh(2r) + \tfrac{3}{4}\sinh(4r)), \quad M_{66}^{\xi} = \tfrac{5}{8} + \cosh(2r) + \tfrac{3}{8}\cosh(4r).$$

## 2. Two-particle jump operators

For two-body jump operators $\{\hat{\ell}_j'\} = \{\hat{a}^2, (\hat{a}^\dagger)^2\}$, we can write the correlation matrix entries as

$$M_{11}'^{\xi} = \tfrac{1}{2}\left(3 - \cos(2\theta) + (1 + \cos(2\theta))\cosh(4r)\right), \quad M_{12}'^{\xi} = \tfrac{1}{2}\sin(2\theta)(\cosh(4r) - 1),$$

$$M_{13}'^{\xi} = -\tfrac{\sqrt{2}}{2}\sin(\theta)\left(\sinh(4r) - \sinh(2r)\right), \quad M_{14}'^{\xi} = M_{15}'^{\xi*} = -\tfrac{i\sqrt{2}}{2}e^{i\theta}\sinh(4r),$$

$$M_{16}'^{\xi} = -\tfrac{\sqrt{2}}{2}\sin(\theta)\left(\sinh(4r) + \sinh(2r)\right),$$

$$M_{22}'^{\xi} = \tfrac{1}{2}\left(3 + \cos(2\theta) + (1 - \cos(2\theta))\cosh(4r)\right), \quad M_{23}'^{\xi} = \tfrac{\sqrt{2}}{2}\cos(\theta)(\sinh(4r) - \sinh(2r)),$$

$$M_{24}'^{\xi} = M_{25}'^{\xi*} = \tfrac{\sqrt{2}}{2}e^{i\theta}\sinh(4r), \quad M_{26}'^{\xi} = \tfrac{\sqrt{2}}{2}\cos(\theta)\left(\sinh(4r) + \sinh(2r)\right),$$

$$M_{33}'^{\xi} = \tfrac{71}{32} - \tfrac{13}{4}\cosh(2r) + \tfrac{3}{2}\cosh(4r) - \tfrac{3}{4}\cosh(6r) + \tfrac{9}{32}\cosh(8r),$$

$$M_{34}'^{\xi} = M_{35}'^{\xi*} = \tfrac{1}{8}e^{2i\theta}(-\tfrac{29}{4} + 3\cosh(2r) + 5\cosh(4r) - 3\cosh(6r) + \tfrac{9}{4}\cosh(8r)), \tag{A2}$$

$$M_{36}'^{\xi} = \tfrac{1}{4}(\tfrac{15}{8} - 3\cosh(4r) + \tfrac{9}{8}\cosh(8r)),$$

$$M_{44}'^{\xi} = M_{55}'^{\xi} = \tfrac{1}{8}(\tfrac{91}{4} + 23\cosh(4r) + \tfrac{9}{4}\cosh(8r)), \quad M_{45}^{\xi} = \tfrac{9}{32}e^{-4i\theta}(3 - 4\cosh(4r) + \cosh(8r)),$$

$$M_{46}'^{\xi} = M_{56}'^{\xi*} = \tfrac{1}{8}e^{-2i\theta}(-\tfrac{29}{4} - 3\cosh(2r) + 5\cosh(4r) + 3\cosh(6r) + \tfrac{9}{4}\cosh(8r)),$$

$$M_{66}'^{\xi} = \tfrac{71}{32} + \tfrac{13}{4}\cosh(2r) + \tfrac{3}{2}\cosh(4r) + \tfrac{3}{4}\cosh(6r) + \tfrac{9}{32}\cosh(8r).$$

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
