# Peer review of "Lindbladian reverse engineering for general non-equilibrium steady states: A scalable null-space approach"

_SciPost Physics_

## Round 1 · Referee Report · Anonymous (Referee 1) · 2024-9-18

Strengths

  1. Originality: The proposed method offers a new approach to reconstructing Lindblad master equations. The method's scalability, with quadratic dependence on the number of jump operators, is a significant improvement compared to methods with exponential growth.

  2. Potential Applications: The method's applicability to a wide range of quantum systems, including those with large Hilbert spaces, suggests its potential for practical use in various fields.

  3. Clear Presentation: The manuscript is well-written and presents the concepts and the mathematical methods clearly and understandably.

Weaknesses

  1. Limited Scope of Examples: The focus on systems with macroscopic parameters limits the generalizability of the method. A broader range of examples would demonstrate its applicability to more complex systems.

  2. Lack of Comparative Analysis: While the authors stress the computational efficiency of their method, a more detailed comparison to existing techniques would make the argument more solid

Report

The authors present a method to reconstruct the form of a Lindblad master equation by reverse engineering using non-equilibrium steady states. Examples of the method based on Gaussian bosonic and collective spin systems are presented. The main result is that the complexity of the problem scales with the square of the number of jump operators, thus avoiding exponential growth with the size of the Hilbert space. The authors also show that their approach can be used as a no-go theorem to detect whether a Lindbladian has a steady state by looking at the absence of null eigenvalues of the correlation matrix (instead of the full Lindbladian).

Comments:

From a theoretical point of view, the manuscript looks very interesting as it offers a clear advantage over some brute-force methods, but the computational advantage over other existing techniques is not discussed in any detail.

The two examples discussed describe the case where the dynamics of the dissipative system can be described by some macroscopic parameters. However, it is not clear how difficult this would be for a general case.

The authors promise to discuss cases beyond the Markovian limit for the master equation. However, the discussion of how the method deals with non-Markovian maps is somewhat lacking.

Minor comment: the word "Lindbladian" is sometimes spelled "Lindbaldian".

In conclusion, I think that the manuscript meets the criteria for acceptance by SciPost Physics. I am favorable toward the publication in SciPost Physics once the authors have adequately addressed the comments presented in my report.

Requested changes

  1. Provide a more quantitative comparison of the computational complexity of the proposed method with other existing techniques.
  2. Include at least the recipe to be followed in an example that does not have simplifying macroscopic parameters.
  3. Discuss the challenges and limitations of applying the method to non-Markovian systems, or diminish what is promised in the manuscript.

Recommendation

Ask for minor revision

---

## Round 1 · Referee Report · Anonymous (Referee 2) · 2024-10-3

Strengths

1- New methods for deriving Lindblad for given steady state 2-Method is potentially simple

Weaknesses

1-Examples are solely for benchmarking the theory

Report

The authors Leonardo da Silva Souza and Fernando Iemini derive in their draft with the title ''Lindbladian reverse engineering for general non-equilibrium steady states: A scalable null-space approach'' a method to calculate the coefficients scaling predefined Hamiltonian terms and jump operators that evolve a quantum system into a given stationary state. The method is quite general and assumes time evolution of density operators with Lindblad master equations. The authors demonstrate that finding the coefficients require to solve a linear system of L x K^2 dimension where L is the number of Hamiltonian terms and K the number of jump operators. They apply this approach to harmonic oscillators where they predict how to generate coherent and squeezed states and to a collective spin model.

My personal opinion is that this draft is very interesting and the method that the authors apply can be useful for many different disciplines working with open quantum systems. With this I argue that this draft is potentially suitable for SciPost Physics. The biggest weakness of this draft are the examples and before I come to several minor criticims I would like the authors to address this major point.

Major: I would have expected that the authors use their developed method to derive new results but for some reason the authors only discuss examples which, I would say, only benchmark their theory. While this is definitely a good check, I would appreciate if the authors could add one new example. Alternatively, I would also be happy if the authors could at least clarify what are the new insights that one gains from these known examples.

Requested changes

Minor:

(i) This is maybe because of my ignorance: it took me a while to understand that the steady state but also the hermitian and jump operators are given. Could the authors clarify this before Eq. (4), where you write the steady state is known. Maybe just add there that also the hermitian and jump operators are know. I guess in an experiment this would mean you have a certain but possibly not full control over the system. Maybe the authors could comment?

(ii) The authors write: ''This function computes roughly the norm [...]'' Why roughly?

(iii) I believe the challenge in the authors' method, which is basically captured by Eq.(9), is the calculation of the elements of the matrix M. This would in general require to calculate complicated operator products. Can the authors comment on this? How does one calculate M for a more general situation where one cannot rely on analytical results as in the examples.

(iv) I did not understand the comment on ''to explore non-Markovian maps''. What does this approach have to do with non-markovian maps? Why is that relevant and how does it connect to the non-positivity of gamma?

(v) I find the comment on ''unique possible solution in Lindblad form'' a bit missleading because there are in my opinion many solutions as soon as one adds a frequency of the harmonic oscillator (h3=a^dag a). Maybe the authors could add a comment.

(vi) The authors write ''Squeezed states could be generated [...]''. I think the authors should write ''can'' instead.

(vii) For a harmonic oscillator with linear driving, dissipation does in general stabilize the state even when driven on resonance. For a parameterically driven oscillator this is not necessarily the case (especially when driven on parametric resonance which the authors study because they have not added the a^dag a term for h). I guess this is what the authors rediscover here (because they find these two possibly unphysical solutions in Eqs.(17)). Is it the case that the two solutions describe an amplification without proper steady state? Also what happens if one adds a finite detuning?

(viii) Related to (vii). Is the motivation of introducing two-particle dissipation that single particle dissipation is insufficient to stabilize the parametric resonance?

(ix) Can the authors remind the reader when they write about Strong dissipative (and weak dissipative) case what they mean by this? Please just repeat \kappa/\omega_0>(<)0.

(x) The authors write ''On the other hand, due to the vanishing of the smallest eigenvalue with system size it suggests that the method should still be feasible for large system sizes''. My question is where this implication comes from? I mean, for instance, is the infinite temperature state (Eq.(23)) for N to infinity even a proper limit?

Recommendation

Ask for minor revision

---

## Round 2 · Referee Report · Anonymous (Referee 2) · 2025-11-25

Report

I find the comments and response of the authors regarding the questions and criticisms from my previous report sufficiently convincing (although there was an unimportant confusion about the previous comments of both Referees). In addition, I like that the authors have added the random local model. I support publication in SciPost Physics.

Recommendation

Publish (meets expectations and criteria for this Journal)

---

## Round 2 · Referee Report · Anonymous (Referee 1) · 2025-12-4

Report

I am satisfied with the answers given by the authors to my previous report. Also, reading the report of the other referee and the answers given, I will recommend the manuscript for publication in its present scientific form. I will just list here a few typos I found (one was already in my previous report):

repeated : Lindbaldian ->Lindbladian
page 1: maximun -> maximum
page 4: steady steady -> steady state

Recommendation

Publish (meets expectations and criteria for this Journal)

---

## Round 2 · Author Response

Dear Editor,

Please find attached the revised manuscript titled ``Lindbladian reverse engineering for general non-equilibrium steady states: A scalable null-space approach''. We are grateful to the reviewers for their time and insightful comments. We have carefully considered all the feedback and detailed our responses below. We hope with these changes, the article is now ready for publication.

Yours Sincerely, Dr. Leonardo da Silva Souza - on behalf of the Authors.

Authors’ Response to Referee 1:

(i) This is maybe because of my ignorance: it took me a while to understand that the steady state but also the hermitian and jump operators are given. Could the authors clarify this before Eq. (4), where you write the steady state is known. Maybe just add there that also the hermitian and jump operators are know. I guess in an experiment this would mean you have a certain but possibly not full control over the system. Maybe the authors could comment?

Response: We thank the referee for this valuable remark. Before Sec. III (Examples), we have included a step-by-step guide outlining the implementation of the method. In Step 1, we now explicitly state that, in addition to the target NESS, one must also define suitable operator bases for both the Hamiltonian (${\hat h_j}$) and the jump operators (${\hat \ell_j}$), chosen to capture the relevant physical configurations of interest. We hope this clarification adequately addresses the referee’s observation regarding the prior knowledge of these operators.

(ii) The authors write: ''This function computes roughly the norm [...]'' Why roughly?

Response: Thank you for the remark. To be more precise, the relation $R(\hat \rho) = Tr(\mathcal{ L}[\hat \rho]^{\dagger}\mathcal{ L}[\hat \rho])$ corresponds to the square of the Frobenius norm. We have revised the sentence to explicitly include this clarification.

(iii) I believe the challenge in the authors' method, which is basically captured by Eq.(9), is the calculation of the elements of the matrix M. This would in general require to calculate complicated operator products. Can the authors comment on this? How does one calculate M for a more general situation where one cannot rely on analytical results as in the examples.

Response: We thank the referee for this insightful comment. Indeed, the main numerical challenge in applying the method lies in computing the elements of the correlation matrix M and determining its kernel (null space). The matrix dimension is $(J+K^2)\times(J+ K^2)$, where $J$ and $K$ are the numbers of elements in the sets of Hermitian and jump-operator bases, respectively. In the collective spin model, both the correlation matrix and its kernel were obtained numerically using standard MATLAB linear-algebra routines. In the revised version, we also include an analysis of a many-body local model—a spin-1/2 chain with random local interactions and single-site jump operators—where all computations were performed numerically without relying on analytical expressions.

It is worth noting that in some of our examples, the numerical effort was dominated by the computation of the NESS of the Lindbladians, as it was required both as input and for comparison between the target NESS and the non-equilibrium steady state generated by the reconstructed Lindbladian. This restriction led us to work with small chains. However, in practical applications of the method, the NESS correlations is already provided as input, so the computational bottleneck associated with obtaining it is removed, allowing one to address significantly larger system sizes.

(iv) I did not understand the comment on ''to explore non-Markovian maps''. What does this approach have to do with non-markovian maps? Why is that relevant and how does it connect to the non-positivity of gamma?

Response: We thank the referee for this pertinent question. We have moderated the claims in the manuscript, leaving such studies as a perspective for future work. Recent studies have proposed using negative decoherence rates, as they appear in the canonical form of the master equation, to characterize non-Markovianity. It would be interesting to explore whether the negative eigenvalues of the dissipative matrix obtained within our framework could be interpreted as signatures of non-Markovian dynamics, corresponding to system–environment interactions that allow partial reversals of earlier dissipative processes.

(v) I find the comment on ''unique possible solution in Lindblad form'' a bit missleading because there are in my opinion many solutions as soon as one adds a frequency of the harmonic oscillator ($h3=a^{\dagger} a$). Maybe the authors could add a comment.

Response: We thank the referee for the comment. It is indeed true that if one expands the operator basis of the Lindbladian, for instance, by including the quadratic term ($h_3 = a^{\dagger}a$), additional solutions may emerge. Our statement regarding the uniqueness of the solution refers specifically to the given operator basis. We have rewritten the sentences preceding this statement to clarify the this information, and we hope that, together with the newly added implementation guidelines, this point is now clearer.

(vi) The authors write ''Squeezed states could be generated [...]''. I think the authors should write ''can'' instead.

Response: We have implemented the change.

(vii) For a harmonic oscillator with linear driving, dissipation does in general stabilize the state even when driven on resonance. For a parameterically driven oscillator this is not necessarily the case (especially when driven on parametric resonance which the authors study because they have not added the $a^dag a$ term for h). I guess this is what the authors rediscover here (because they find these two possibly unphysical solutions in Eqs.($17$)). Is it the case that the two solutions describe an amplification without proper steady state? Also what happens if one adds a finite detuning?

Response: In general, our goal was to illustrate the method rather than to analyze the complete physical landscape of the solutions. When the operator basis is expanded by adding $\hat{h}_{3}=\hat{a}^{\dagger}\hat{a}$, an additional solution arises, in addition to the three discussed previously, and this new solution also exhibits a non-positive dissipative matrix. Depending on how these four solutions are combined, one may obtain a purely Hamiltonian solution; however, no completely positive Markovian solution satisfying $[\hat{H},\hat{L}]\neq 0$ is generated. The study of solutions with a non-positive dissipative matrix may nevertheless be of interest. As discussed earlier, future works could investigate whether such solutions can still preserve physicality, with negative decoherence rates potentially serving as signatures of non-Markovian behavior.

(viii) Related to (vii). Is the motivation of introducing two-particle dissipation that single particle dissipation is insufficient to stabilize the parametric resonance?

Response: We thank the referee for this question. The motivation for introducing two-particle dissipation, rather than single-particle dissipation, was to construct a Lindbladian in which the interplay between the Hamiltonian and more complex dissipative terms ($[\hat H,\hat L]\neq 0$) that are capable of generating the squeezed vacuum state as a non-equilibrium steady state. Given the structure of the squeezed states (Eq.$16$) - as exponential of two-particle operators - a natural candidate follows by considering two-particle dissipative channels.

(ix) Can the authors remind the reader when they write about Strong dissipative (and weak dissipative) case what they mean by this? Please just repeat $\kappa/\omega_0>(<)0$.

Response: We have implemented the change.

(x) The authors write ''On the other hand, due to the vanishing of the smallest eigenvalue with system size it suggests that the method should still be feasible for large system sizes''. My question is where this implication comes from? I mean, for instance, is the infinite temperature state (Eq.(23)) for N to infinity even a proper limit?

Response: We thank the referee for pointing this out. Equation (23) in the previous version contained a typo: the infinite-temperature state was written without normalization, although all computations were performed assuming the normalized form. This has been corrected in the revised manuscript (now Eq. (26)). With this correction, the finite-size analysis remains valid, indicating that even under perturbations, for sufficiently large chains, the method still yields a consistent solution.

Authors’ Response to Referee 2:

Referee major coment: My personal opinion is that this draft is very interesting and the method that the authors apply can be useful for many different disciplines working with open quantum systems. With this I argue that this draft is potentially suitable for SciPost Physics. The biggest weakness of this draft are the examples and before I come to several minor criticims I would like the authors to address this major point. Major: I would have expected that the authors use their eveloped method to derive new results but for some reason the authors only discuss examples which, I would say, only benchmark their theory. While this is definitely a good check, I would appreciate if the authors could add one new example. Alternatively, I would also be happy if the authors could at least clarify what are the new insights that one gains from these known examples.

Response: We thank the Referee for their positive assessment of our work and for the insightful comments that have helped us improve the manuscript. The Referee points out that the current examples primarily serve to benchmark our new reverse-engineering method. We agree that applying the method to derive fundamentally new physical results is an exciting prospect. We respectfully wish to clarify, however, that the primary novel contribution of this work is the development of the method itself — (i) its formal derivation, (ii) the analysis of its properties and subtleties, and (iii) the demonstration of its practical application across different benchmark systems. We believe that establishing a robust and well-understood theoretical tool is a significant result in its own right, forming an essential foundation upon which future discoveries can be built. It would be therefore unfair the requirement of having it all (i)-(iii) + deriving new results (as we understood (i)-(iii) are not the ones mentioned here by the referee) in this single work. In the updated version we have also added a new example for another class of models, namely with short-range interactions, as well as expanded the discussion on the methodology of the method, providing a step-by-step guide on how to use the method in practice and highlighting its peculiarities when dealing with the form of solutions.

Nevertheless, sharing with the Referee's vision for the method's potential, we have expanded the concluding section (see "Another promising applications of our method is that...") to highlight what we believe to be a promising application of our method: finding alternative Lindbladians for a given target NESS. Specifically, given a NESS generated by a specific Lindbladian $\mathcal{L}_{\rm full}$, one may argue if this same state could be engineered by an alternative Lindbladian $\mathcal{L}{\rm alt} \neq \mathcal{L}$. The alternative Lindbladian could potentially be more experimentally feasible, bringing the preparation and use of the NESS closer to the realms of practicality.

Our preliminary studies, which we describe in the manuscript, are encouraging. For instance, in a study of the random spin model and its corresponding input NESS ($\hat \rho_{exact, NESS}$), we used our method to reverse-engineer a Lindbladian under a constrained ansatz: we forbid dissipative channels at certain sites but allowed for longer-range Hamiltonian interactions. The method successfully found an alternative Lindbladian whose steady state had a fidelity of $\approx 99\%$ with the original target state ($\hat \rho_{exact, NESS}$). The obtained reverse engineered Lindbladian dealt with the lack of dissipation by exploiting the Hamiltonian interactions in a longer-range, thereby recovering the same class of NESS. This suggests that a desired complex non-equilibrium steady state (NESS) could potentially be engineered by a different, and possibly more experimentally feasible, set of controls — for example, trading challenging local dissipation for tunable long-range interactions.

While a comprehensive study of this application is beyond the scope of the present work (which focuses on introducing and validating the method), we have revised the text to make it clear that enabling such routes is a consequence of our results.

(i) Provide a more quantitative comparison of the computational complexity of the proposed method with other existing techniques.

Response: We appreciate the referee’s concern. Indeed, a quantitative comparison of computational complexity would be valuable, and we had considered including such a discussion in an earlier version of the manuscript. However, a quantitative comparison between our method and all existing approaches is not always possible or strictly direct, as the various methods differ significantly in scope, objectives, and underlying assumptions. A detailed comparison could therefore be both unfeasible and potentially misleading. Instead, in the main text, we have chosen to present the core principles of each class of methods, allowing the reader to understand the broader landscape of available strategies. The main challenges preventing a direct quantitative comparison are summarized below:

  1. Different classes of Lindbladians: the various methods are designed to handle different classes of problems. For example, analytical techniques are often employed to restricted set of Lindbaldians (e.g, Gaussian maps, those satisfying detailed balance) for which for which they are efficient. In contrast, full tomography can handle general Lindbladians but becomes computationally prohibitive, as its complexity scales poorly with system size.

  2. Different Specific Goals: while full tomography aims to reconstruct the entire quantum channel for a fixed finite-time evolution ($t_0 \to t_f$), variational methods typically aim to find a Lindbladian that best fits the dynamics over a continuous time interval $[0, T]$. Our method on the other hand is specifically designed to determine the generator that drives any initial state toward the correct Non-Equilibrium Steady State (NESS), focusing on the asymptotic limit ($t \to \infty$). These fundamentally different goals make it difficult to situate all approaches within a single computational-complexity framework.

  3. Exact vs approximate nature: while full tomography and our method are exact (up to state preparation and measurement errors), variational methods are inherently approximate, seeking an $\epsilon$-close solution. Their computational cost is a function of this freely tunable parameter $\epsilon$ and depends heavily on optimization details (e.g., in neural networks it depends on the network architecture, training time, learning rates). This introduces a free variable that makes a strict, direct complexity comparison with exact methods unfeasible.

In summary, due to the differing applicable classes, fundamental goals, and the exact/approximate nature of the various techniques, a general and direct quantitative complexity analysis is not straightforward. Our approach in the manuscript has been to provide a qualitative comparison that highlights the trade-offs: (1) analytical methods - good, but for a restrict class -, (2) full tomography - valid for general maps but computationally inefficient - and (3) variational methods - work also for general models but with the drawback of not being exact and at the cost of minimizing a non-linear cost function. We believe that this discussion equips the reader with the necessary context to understand the position and utility of our proposed reverse-engineering technique within the broader toolkit.

(ii) Include at least the recipe to be followed in an example that does not have simplifying macroscopic parameters.

Response: We thank the referee for this valuable remark. In the revised version, we have added a step-by-step guide before Sec. III (Examples), outlining the implementation of the method. In addition, we now include an analysis of a many-body local model, a spin-1/2 chain with random local interactions and single-site jump operators, which serves as an explicit example without simplifying macroscopic parameters.

(iii) Discuss the challenges and limitations of applying the method to non-Markovian systems, or diminish what is promised in the manuscript.

We appreciate the referee’s constructive comment. In the revised version, we have moderated the discussion regarding the possible extension of the method to non-Markovian regimes, and the potential connection between non-positive $\hat{\gamma}$ and non-Markovian behavior is now presented only as a perspective for future work.

---

## Round 2 · List of Changes

The modifications made to the manuscript are highlighted in blue. The main changes are as follows:

• An additional affiliation has been included for author Leonardo da Silva Souza:
Instituto Federal de Educação, Ciência e Tecnologia de Mato Grosso, Av. Rio Grande do Sul, 2131, St. Industrial, 78640-000 Canarana, Mato Grosso, Brazil.

• A step-by-step guide detailing the implementation of the method has been added to Sec. II.

• An analysis of a many-body local model—a spin-1/2 chain with random local interactions and single-site jump operators—has been incorporated to Sec. III and IV.

• The subsection “Robustness of the method” has been promoted to Section IV, and the robustness analysis has been extended to include the Random Local Model.

• The discussion concerning the possible extension of the method to non-Markovian regimes has been moderated and clarified in Sec. II and in the Conclusion.

• The conclusion now includes a discussion of future perspectives motivated by our findings.

• A new reference [29], Control of open quantum systems via dynamical invariants, has been added.

---

## Editorial Decision

accepted_in_target_journal